# Laser Biostimulation Induces Wound Healing-Promoter β2-Defensin Expression in Human Keratinocytes via Oxidative Stress

**DOI:** 10.3390/antiox12081550

**Published:** 2023-08-03

**Authors:** Mario Migliario, Preetham Yerra, Sarah Gino, Maurizio Sabbatini, Filippo Renò

**Affiliations:** 1Traslational Medicine Department, Università del Piemonte Orientale, Via Solaroli n. 17, 28100 Novara, Italy; mario.migliario@med.uniupo.it; 2Health Sciences Department, Università del Piemonte Orientale, Via Solaroli n. 17, 28100 Novara, Italy; 20035225@studenti.uniupo.it (P.Y.); sarah.gino@uniupo.it (S.G.); 3Sciences and Innovative Technology Department, Università del Piemonte Orientale, Viale T. Michel 11, 15121 Alessandria, Italy; maurizio.sabbatini@uniupo.it

**Keywords:** laser biostimulation, innate immunity, hBD-1, hBD-2, oxidative stress, wound healing

## Abstract

The innate immune system is the first line of defense of the body composed of anatomical barriers, such as skin and mucosa, as well as effector cells, antimicrobial peptides, soluble mediators, and cell receptors able to detect and destroy viruses and bacteria and to sense trauma and wounds to initiate repair. The human β-defensins belong to a family of antimicrobial small cationic peptides produced by epithelial cells, and show immunomodulatory and pro-healing activities. Laser biostimulation is a therapy widely used to contrast microbial infection and to accelerate wound healing through biological mechanisms that include the creation of oxidative stress. In this paper, we explored laser biostimulation’s ability to modulate the production of two β-defensins, hBD-1 and hBD-2, in human keratinocytes and whether this modulation was, at least in part, oxidative-stress-dependent. Human spontaneously immortalized keratinocytes (HaCaT) were stimulated using laser irradiation at a 980 nm wavelength, setting the power output to 1 W (649.35 mW/cm^2^) in the continuous mode. Cells were irradiated for 0 (negative control), 5, 10, 25 and 50 s, corresponding to an energy stimulation of 0, 5, 10, 25 and 50 J. Positive control cells were treated with lipopolysaccharide (LPS, 200 ng/mL). After 6 and 24 h of treatment, the cell conditioned medium was collected and analyzed via ELISA assay for the production of hBD-1 and hBD-2. In another set of experiments, HaCaT were pre-incubated for 45 min with antioxidant drugs—vitamin C (Vit. C, 100 µM), sodium azide (NaN_3_, 1 mM); ω-nitro-L-arginine methyl ester (L-NAME, 10 mM) and sodium pyruvate (NaPyr, 100 µM)—and then biostimulated for 0 or 50 s. After 6 h, the conditioned medium was collected and used for the ELISA analysis. The hBD-1 and hBD-2 production by HaCaT was significantly increased by single laser biostimulation after 6 h in an energy-dependent fashion compared to basal levels, and both reached production levels induced by LPS. After 24 h, only hBD-2 production induced by laser biostimulation was further increased, while the basal and stimulated hBD-1 levels were comparable. Pre-incubation with antioxidative drugs was able to completely abrogate the laser-induced production of both hBD-1 and hBD-2 after 6 h, with the exception of hBD-1 production in samples stimulated after NaN_3_ pre-incubation. A single laser biostimulation induced the oxidative-stress-dependent production of both hBD-1 and hBD-2 in human keratinocytes. In particular, the pro-healing hBD-2 level was almost three times higher than the baseline level and lasted for 24 h. These findings increase our knowledge about the positive effects of laser biostimulation on wound healing.

## 1. Introduction

Defensins represent the largest family of antimicrobial small cationic peptides found in various organisms, including humans. These peptides play a vital role in the innate immune response by providing defense against invading pathogens. Defensins are characterized by their unique structural features, including a primary chain of 29–35 amino acids, a variable quantity of arginine, and the presence of six cysteine residues that form three intramolecular disulfide bridges. These disulfide bridges contribute to the stabilization of a three-dimensional β-sheet structure [1].

Defensins are believed to promote the activation of the immune response, following infections, through a chemotactic effect on monocytes [2]. The two major categories of defensins are α-defensins and β-defensins. α-Defensins are initially produced as propeptides and are activated through proteolytic cleavage, typically by trypsin. In humans, α-defensins are predominantly expressed in neutrophils and Paneth cells, specialized cells found in the intestine. Neutrophil-derived α-defensins provide a crucial defense mechanism at sites of infection, while Paneth cell-derived α-defensins protect the intestinal mucosa against microbial invasion [3].

On the other hand, β-defensins are represented by three distinct peptides, β-defensin 1, 2, and 3, and they are found in epithelial cells [4].

The β-defensins’ functional role appears not to be limited to antimicrobial activity, as they showed immunomodulatory activity, providing a link between innate and adaptive immunity and regulating immune responses, including the recruitment and activation of immune cells, such as monocytes and dendritic cells [5].

Human β-defensin 1 (hBD-1) was first discovered by analyzing large amounts of hemofiltrate and it is constitutively expressed [6]. Human β-defensin 2 (hBD-2) is primarily found in epithelial cells, particularly keratinocytes, which suggests a possible role in defending the large surface areas of the integumentary system [7,8]. Human β-defensin 3 (hBD-3) was first isolated from human lesional psoriatic scales, from primary keratinocytes and lung epithelial cells pretreated with *P. aeruginosa* [9,10,11].

The induction of β-defensins is not solely dependent on Toll-like receptors (TLRs), which recognize and bind pathogen-associated molecular patterns (PAMPs) [12]. TLR activation triggers downstream signaling pathways, such as MAPK or NF-κB, leading to a proinflammatory response characterized by the secretion of cytokines, chemokines, and β-defensins. However, independent pathways involving NOD2, IL-17R, and PAR-2 can also induce β-defensin expression [12].

Laser biostimulation, a therapeutic technique widely used to combat bacterial infections, reduce pain and inflammation, and accelerate tissue repair, has received significant attention in recent years [13]. This non-invasive treatment modality utilizes specific wavelengths and exposure times to alter cellular behavior without causing significant heating. Laser biostimulation typically employs light within the range of 390 to 1100 nm, with variations depending on the targeted tissue depth. The light can be delivered in continuous or pulsed waves, with low frequencies ranging from 0.04 to 50 J/cm^2^ [13].

Although the precise mechanisms underlying the effects of laser biostimulation are not fully understood, there is increasing recognition of the role of oxidative stress in mediating these effects. It is believed that red and infrared (NIR) light is absorbed by chromophores, such as cytochrome c oxidase (CCO) in the mitochondria, and potentially by photoacceptors in the cell’s plasma membrane [14]. This absorption leads to the generation of reactive oxygen species (ROS) [15,16]. ROS, in turn, can activate various intracellular signaling pathways, modulate the affinity of transcription factors, and influence processes such as proliferation, cell survival, tissue repair, regeneration, and inflammation [13,17,18,19,20,21].

Emerging evidence suggests that laser biostimulation-induced ROS play a crucial role in promoting cell proliferation, stimulating the release of cytokines and nitric oxide (NO), and even triggering netosis, an innate immune mechanism involving the release of antimicrobial components from neutrophils [17,22,23]. Given these intriguing findings, we became interested in investigating whether laser biostimulation could induce the production of hBD-1 and hBD-2 in human keratinocytes, and whether this stimulation is dependent on the induction of oxidative stress.

## 2. Materials and Methods

### 2.1. Cell Culture

The human spontaneously immortalized keratinocyte cell line HaCaT was used for all experiments [24]. The cells were obtained from Cell Lines Service GmbH (Eppelheim, Germany™) and cultured in 25 cm^2^ plastic flasks. Dulbecco’s modified Eagle medium (DMEM) with high glucose levels, supplemented with 10% heat-inactivated fetal bovine serum (FBS) and 1% penicillin-streptomycin (all from Immunological Sciences, Rome, Italy), was used as the culture medium. The cells were incubated in a humidified incubator at 37 °C with 5% CO_2_. Phosphate-buffered saline (PBS 1X, pH = 7.2) was used for sterilization and experimental procedures. HaCaT cells were detached using trypsin (0.25% trypsin in PBS containing 0.05% EDTA), resuspended, and seeded in 24-well plates at a density of 1 × 10^5^ cells per well in complete DMEM. The cells were allowed to adhere and grow until they reached 100% confluence. Once confluent, the cells were washed twice with PBS 1X and then cultivated in serum-free DMEM for 24 h at 37 °C with 5% CO_2_.

### 2.2. In Vitro Laser Biomodulation

HaCaT cells were stimulated using the DMT Giotto laser equipment (DMT srl, Lissone, Italy). Laser irradiation was performed at a wavelength of 980 nm using a 600 μm optical fiber. The power output was set to 1 W (649.35 mW/cm^2^), as previously described [22]. Laser stimulation was carried out in continuous mode, with the light source positioned vertically above each well at a distance of 9.7 cm from the bottom of the well, ensuring complete irradiation of the well area (1.54 cm^2^). Before laser irradiation, 300 μL of serum-free DMEM without phenol red was added to each well. The plates were uncovered, and laser stimulation was applied for 0 s (negative control), 5, 10, 25, or 50 s, corresponding to energy stimulations of 0, 5, 10, 25, and 50 J, respectively, and energy densities (spatial average energy fluence) of 0, 3.25, 6.5, 16.23, and 32.47 J/cm^2^, respectively. Following laser stimulation, the HaCaT cells were maintained in 300 μL of serum-free DMEM without phenol red in a humidified atmosphere with 5% CO_2_ at 37 °C for either 6 or 24 h. Positive control wells were treated with lipopolysaccharide (LPS) at a concentration of 200 ng/mL. The LPS used was purchased from Sigma-Aldrich (Saint Louis, MO, USA) and obtained from Escherichia coli 0111:B4. After 6 and 24 h of treatment, the conditioned media from the cells in the different experimental conditions were collected and analyzed using ELISA to measure the production of hBD-1 and hBD-2.

### 2.3. Role of Oxidative Stress in Laser Biostimulation

To evaluate the role of oxidative stress in the laser biostimulation of hBD-1 and hBD-2 production, a second set of experiments was conducted. HaCaT cells were pre-incubated for 45 min with specific compounds to counteract oxidative stress. The compounds used were: vitamin C (Vit. C, 100 µM) to counteract the formation of superoxide anion, H_2_O_2_, hypochlorite, and hydroxyl radicals; sodium azide (NaN_3_, 1 mM), which interferes with cellular respiration by limiting the use of oxygen; ω-nitro-L-arginine methyl ester (L-NAME, 10 mM), which reduces nitric oxide production; and sodium pyruvate (NaPyr, 100 µM). Half of the samples were then subjected to biostimulation at an intensity of 50 J. After 6 h of treatment, the conditioned media from the cells in the various experimental conditions were collected and used for ELISA analysis of hBD-1 and hBD-2 production.

### 2.4. Analysis of hBD-1 and hBD-2 Production

The concentrations of hBD-1 and hBD-2 were measured in the conditioned medium of HaCaT cells using commercially validated Human Beta Defensin 1 and 2 ELISA kits (Immunological Sciences, Rome, Italy), respectively. The IK5164 Hu Beta1 Defensins ELISA Kit had a sensitivity range of 0–18.75 pg/mL, and the IK5165 Hu Beta2 Defensins ELISA Kit had a sensitivity range of 0–37.5 pg/mL. Each assay was performed in triplicate according to the manufacturer’s instructions. The optical density (O.D.) absorbance of the samples at 450 nm was measured using a microplate reader (Victor X4, Perkin-Elmer, Waltham, MA, USA). The data were expressed as hBD-1 and hBD-2 concentrations in pg/mL of conditioned medium.

### 2.5. Statistical Analysis

Each experiment was conducted in triplicate to ensure statistical significance. Statistical analysis was performed using one-way ANOVA followed by Bonferroni’s post hoc tests. The Prism 4.0 statistical software (GraphPad Software Inc., Boston, MA, USA) was used for the analysis. Probability values of *p* < 0.05 were considered statistically significant.

## 3. Results

The study aimed to investigate the production of hBD-1 and hBD-2 by HaCaT cells following laser stimulation at different time points and energy levels. Commercially available ELISA kits were employed to measure the levels of these defensins in the cell conditioned medium after 6 and 24 h. Figure 1 illustrates the results obtained after 6 h of laser stimulation. Initially, the basal levels of hBD-1 and hBD-2 were found to be approximately equal, with concentrations of 3.6 ± 0.4 pg/mL and 3.6 ± 0.3 pg/mL, respectively. However, when the cells were stimulated with LPS, a physiological stimulant, there was a significant increase in the release of both hBD-1 and hBD-2, reaching concentrations of 6.6 ± 0.8 pg/mL and 11.9 ± 2.4 pg/mL, respectively. Notably, the hBD-2 release induced by LPS was 327 ± 67% of the basal level, while hBD-1 exhibited a comparatively lower increase of 184 ± 23% under the same conditions. Furthermore, laser biostimulation was also able to enhance the release of both hBD-1 and hBD-2 from the keratinocytes after 6 h of stimulation (Figure 1). This effect was found to be dependent on the energy level used. Specifically, hBD-1 concentration increased significantly compared to the basal level following single stimulations of 5 J (4.5 ± 0.3 pg/mL, *p* < 0.05) and 10 J (4.3 ± 0.2 pg/mL, *p* < 0.05). Moreover, higher energy levels of 25 J (5.6 ± 0.5 pg/mL, *p* < 0.001) and 50 J (5.6 ± 0.6 pg/mL, *p* < 0.001) resulted in a further increase in hBD-1 release, with the latter energy level inducing a release comparable to that seen with LPS stimulation. Similarly, the concentration of hBD-2 also exhibited a significant increase compared to the basal level after 6 h of stimulation with 5 J (4.7 ± 0.6 pg/mL, *p* < 0.05), and a rapid increase following 10 J (6.5 ± 0.2 pg/mL, *p* < 0.001), 25 J (12.9 ± 1.2 pg/mL, *p* < 0.001), and 50 J (11.2 ± 1.4 pg/mL, *p* < 0.001) laser stimulation. Notably, the hBD-2 concentration reached levels comparable to those induced by LPS. The results further highlighted the disparity between the two defensins’ responses to laser stimulation, with hBD-1 levels ranging from 125 ± 8% (5 J) to 156 ± 17% (50 J) of the baseline, while hBD-2 concentrations showed a wider range of increase, varying between 131 ± 17% (5 J) and 311 ± 39% (50 J) of the baseline.

To evaluate the long-term effects of laser stimulation, hBD-1 and hBD-2 production was measured after 24 h (Figure 2). The basal levels of both defensins significantly increased at this time point, with hBD-1 reaching 6.9 ± 1.3 pg/mL and hBD-2 reaching 12.8 ± 1.4 pg/mL. Interestingly, while a 24 h LPS stimulation did not further increase hBD-1 production compared to the basal level (6.4 ± 0.4 pg/mL), it still induced a significant increase in hBD-2 release (15.4 ± 0.9 pg/mL, *p* < 0.001). In contrast, laser biostimulation did not result in a further increase in hBD-1 release compared to the basal level at any energy level used after 24 h. However, hBD-2 levels continued to increase compared to the basal level at all energy levels, albeit without an energy-level-dependent effect. Specifically, hBD-2 levels measured in samples stimulated with 5 J (17.3 ± 0.8 pg/mL) and 50 J (15.7 ± 1.0 pg/mL) were not statistically different (Figure 2).

Based on the findings obtained at the 6 h time point after laser stimulation, the researchers decided to investigate the effect of different antioxidant drugs on the production of hBD-1 and hBD-2. They focused on the 50 J energy level and assessed the response at 6 h (Figure 3 and Figure 4). Figure 3 demonstrates that the antioxidant drugs did not alter the basal level of hBD-1; however, almost every drug halved the hBD-1 concentration induced by 50 J laser biostimulation. Notably, vitamin C (Vit. C), L-NAME, and NaPyr reduced the concentration of hBD-1 induced by 50 J laser stimulation (5.6 ± 1.1 pg/mL) to 2.5 ± 0.9 pg/mL, 2.1 ± 0.5 pg/mL, and 2.4 ± 0.4 pg/mL, respectively. In contrast, NaN_3_ failed to reduce the laser-induced increase in hBD-1 (4.8 ± 1.4 pg/mL).

In contrast, Figure 4 presents a different scenario for hBD-2 levels in the presence of antioxidant drugs. Initially, the antioxidant drugs slightly increased the basal level of hBD-2. Specifically, NaN_3_ (5.9 ± 1.4 pg/mL, *p* < 0.05) and L-NAME (5.8 ± 1.1 pg/mL, *p* < 0.05) showed significantly different levels compared to the basal level (3.6 ± 0.3 pg/mL), while Vit. C (5.6 ± 2.3 pg/mL) and NaPyr (4.6 ± 1.3 pg/mL) did not reach statistical significance. However, all the antioxidant drugs effectively inhibited the increased concentration of hBD-2 induced by 50 J laser biostimulation. Notably, the concentration of hBD-2 induced by 50 J laser stimulation (11.2 ± 1.4 pg/mL) was significantly reduced by Vit. C, NaN_3_, L-NAME, and NaPyr to 2.9 ± 1.8 pg/mL, 2.0 ± 1.1 pg/mL, 1.8 ± 0.7 pg/mL, and 2.9 ± 0.3 pg/mL, respectively.

In summary, the results demonstrate that laser biostimulation increased the release of hBD-1 and hBD-2 from human keratinocytes in an energy-dependent manner. While hBD-1 exhibited a modest increase compared to the basal level, hBD-2 showed a more significant and pronounced response. Moreover, antioxidant drugs were found to modulate the production of hBD-1 and hBD-2, with varying effects observed depending on the specific defensin and drug employed.

## 4. Discussion

Laser biomodulation therapy has been widely used for therapeutic purposes to produce analgesia, to modulate immune response and to accelerate wound healing and tissue regeneration [17,22,23,25].

The action mechanism of laser biomodulation seems to be linked to photoacceptors’ molecular configuration being modified by photon absorption, causing a modification of different signaling pathways [26,27] due to energy generation and reactive oxygen species (ROS) production [28]. In fact, it has been demonstrated that laser biomodulation activates nuclear factor-κB (NF-κB) transcription factor, mitogen-activated protein kinases (MAPKs) and growth factors (e.g., transforming growth factor-β1) [29,30]. We have also observed that photobiostimulation is able to increase the activity of granulocytes within the innate immunity [17]. In this report, we explored the hypothesis that laser biostimulation was able to stimulate other components of innate immunity, in particular the production and release of two antimicrobial peptides, hBD-1 and hBD-2, in epithelial cells. We used human spontaneously immortalized keratinocytes (human adult low-calcium high-temperature, HaCaT) that have been extensively used to study the epidermal homeostasis and its pathophysiology, and that express both defensins in response to physical and pro-inflammatory stimuli [7]. In this cellular model, laser biostimulation with an energy range between 5 and 50 J (fluence range = 3.25–32.47 J/cm^2^) was able to induce the production and release of both defensins after 6 h in an energy-dependent fashion compared to unstimulated cells. This single stimulation effect increased after 24 h only for hBD-2, which is considered an inducible β-defensin [8], while disappeared for constitutively expressed hBD-1 [6]. As stated before, ROS production is a key factor in biological effects induced by laser stimulation, and therefore the effect of cell pre-incubation with different antioxidant drugs on laser-induced hBD-1 and hBD-2 was tested after 6 h of 50 J stimulation, an experimental condition during which both β-defensins were induced. We observed a differentiated action of the antioxidants against the two different β-defensins. In the case of hBD-1, the antioxidants did not alter its basal levels, but they abolished the increased production induced by laser stimulation, except in the case of NaN_3_, a mitochondrial cytochrome c oxidase (CCO) inhibitor [31]. Therefore, we could speculate that hBD-1 biostimulation was not mediated by mitochondrial activity. On the other hand, baseline levels of inducible hBD-2 were increased by antioxidant agents, especially in the case of NaN_3_ and L-NAME, a well-known nitric oxide synthase (NOS) inhibitor, while all antioxidant drugs completely abrogated the important laser-induced production of hBD-2. Our data may describe a finely tuned redox-dependent hBD-2 that needs further investigation to be fully elucidated. However, we demonstrated that a single laser biostimulation is capable of inducing a small and short-lived production of hBD-1 and a strong and sustained production of hBD-2. The latter phenomenon could cause two different effects in biostimulation therapy of an epithelium. Firstly, greater protection can be provided by an important effector of innate immunity, and this could be of extreme interest and efficacy in the case of wounds of various origins, which could be subjected to microbial/fungal action. The induction of hBD-2 production in the absence of bacterial stimulation and under controlled and safe conditions can create a very preventive defense in the case, for example, of minor surgical interventions or even in the presence of metabolic dysfunctions (e.g., diabetes).

Second, in addition to its antimicrobial activity, hBD-2 has been implicated in many physiological conditions associated with wound healing. Indeed, hBD-2 increases the production of proinflammatory cytokines and chemokines from keratinocytes and stimulates their proliferation and migration [32], regulates the proliferation, migration and tube formation of human umbilical vein endothelial cells [33] and its cutaneous expression is induced by skin lesions [34,35]. Finally, it has been suggested that inadequate upregulation of hBD-2 plays a pathogenic role in diabetic wounds [36]. Interestingly, accelerated healing of burn wounds with laser biomodulation therapy involves the activation of endogenous latent tumor growth factor β1 (TGF-β1) [37], an important participant during the wound healing process [38] also expressed in keratinocytes [39]. The intriguing point is that laser activation of latent TGF-β1 is redox-dependent [30], and that hBD-2 expression is induced by latent laser-activated TGF-β1 [40]. It is therefore reasonable to imagine that biostimulation therapy can induce the expression of hBD-2 through two overlapping mechanisms: the first acting directly through the production of ROS and the second acting through the activation of TGF-β1 mediated by the ROS action (Figure 5). This second mechanism could also explain why the production of hBD-2, induced by a single laser stimulation, is able to last for at least 24 h.

## 5. Conclusions

Our findings reveal a novel observation regarding the ability of laser biostimulation to induce the production of hBD-1 and hBD-2 in human keratinocytes through redox-dependent mechanisms. Notably, a single laser treatment within the energy range of 5–50 J demonstrated a significant increase in the levels of hBD-1 and hBD-2 compared to baseline. In particular, the levels of the pro-healing hBD-2 induced by the applied energy peaks (10–50 J) were nearly three times higher than the baseline level and maintained this heightened production for at least 24 h. However, it is important to acknowledge certain limitations in our study. Firstly, the use of spontaneously immortalized cell lines, such as HaCaT cells, may not fully represent the complex cellular dynamics and responses observed in primary cells or in vivo conditions. Therefore, future investigations should focus on examining the effects of laser biostimulation on hBD-2 production in dermal fibroblasts and primary cells, including those derived from individuals with diabetes. This would provide a more comprehensive understanding of the therapeutic potential of laser treatment in different cell type and disease contexts. Secondly, our study only involved a single biostimulation session. Exploring the effects of multiple laser treatments, optimized treatment protocols, and variations in laser parameters could provide valuable insights into the kinetics and long-term effects of hBD-1 and hBD-2 production. Furthermore, investigating the impact of laser biostimulation in appropriate in vivo wound models would be essential to validate our findings and establish a mechanistic rationale for the clinical application of lasers in the management of challenging-to-heal wounds. By addressing these limitations through future research endeavors, we aim to advance the field’s understanding of laser biostimulation and its potential for optimizing clinical management strategies in the treatment of chronic, non-healing wounds.

## Figures and Tables

**Figure 1 antioxidants-12-01550-f001:**
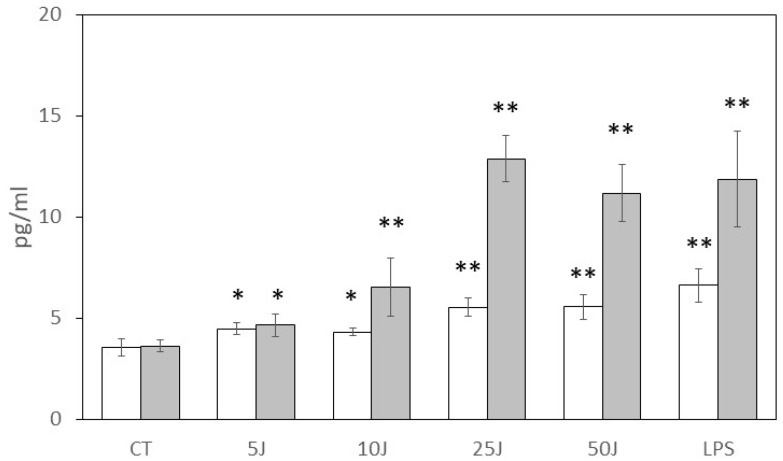
Effect of HaCaT laser biostimulation on hBD-1 (white bars) and hBD-2 (gray bars) production after 6 h. HaCaT were untreated (CT, negative control), treated with different laser energies (5–50 J) or with LPS (200 ng/mL, positive control). Data were obtained from different experiments (*n* = 3) performed in triplicate for each experimental condition. Results are expressed as average pg/mL ± standard deviation (S.D.) of h-BD1 or h-BD2 detected via ELISA assay in the cell conditioned medium after 6 h; * *p* < 0.05; ** *p* < 0.001 compared to unstimulated (CT) samples.

**Figure 2 antioxidants-12-01550-f002:**
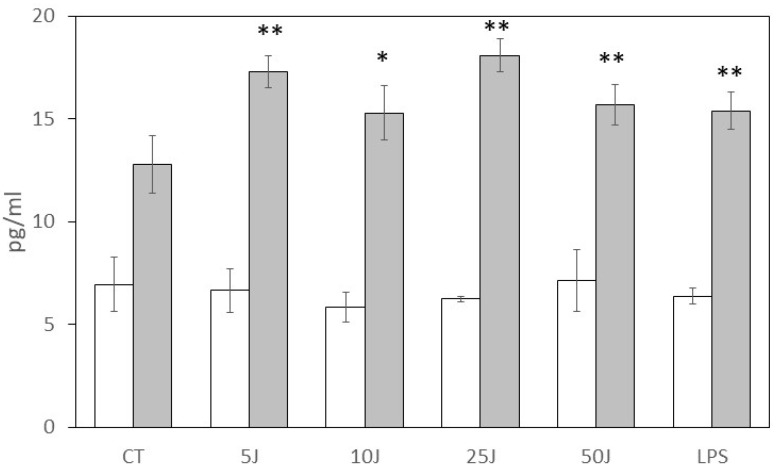
Effect of HaCaT laser biostimulation on h-BD1 (white bars) and hBD-2 (gray bars) production after 24 h. HaCaT were untreated (CT, negative control), treated with different laser energies (5–50 J) or with LPS (200 ng/mL, positive control). Data were obtained from different experiments (*n* = 3) performed in triplicate for each experimental condition. Results are expressed as average pg/mL ± standard deviation (S.D.) of h-BD1 or h-BD2 detected via ELISA assay in the cell conditioned medium after 24 h; * *p* < 0.05; ** *p* < 0.001 compared to unstimulated (CT) samples.

**Figure 3 antioxidants-12-01550-f003:**
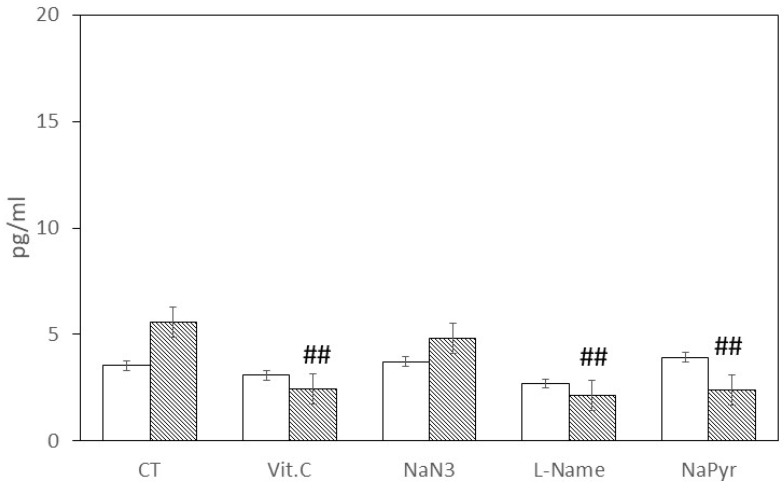
Effect of antioxidant agents on h-BD1 production in HaCaT in basal (unstimulated) conditions (white bars) and under 50 J laser biostimulation (dotted bars) after 6 h. HaCaT were pre-incubated for 45 min with vitamin C (Vit. C, 100 µM), sodium azide (NaN_3_, 1 mM), ω-nitro-L-arginine methyl ester (L-NAME, 10 mM) and sodium pyruvate (NaPyr, 100 µM) and then were left untreated (white bars) or treated with 50 J laser energy (dotted bars). Control (CT) samples were not pre-incubated with antioxidant agents. Data were obtained from different experiments (*n* = 3) performed in triplicate for each experimental condition. Results are expressed as average pg/mL ± standard deviation (S.D.) of h-BD1 or h-BD2 detected via ELISA assay in the cell conditioned medium after 6 h; ## *p* < 0.001 compared to 50 J laser-treated CT samples.

**Figure 4 antioxidants-12-01550-f004:**
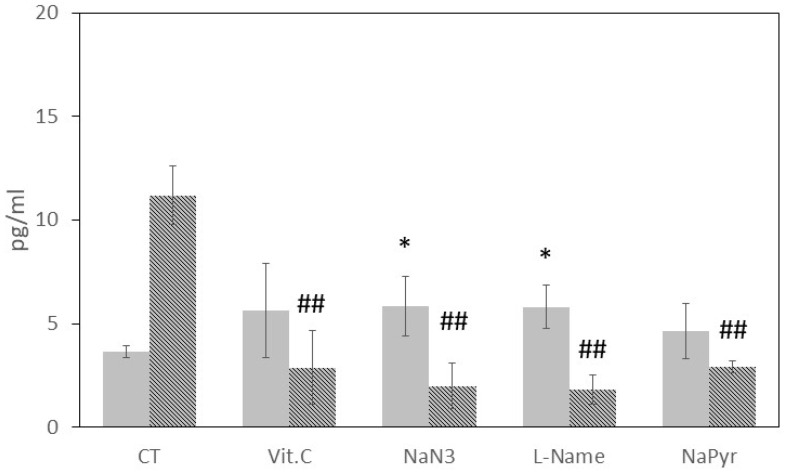
Effect of antioxidant agents on h-BD2 production in HaCaT in basal (unstimulated) conditions (gray bars) and under 50 J laser biostimulation (dotted bars) after 6 h. HaCaT were pre-incubated for 45 min with vitamin C (Vit. C, 100 µM), sodium azide (NaN_3_, 1 mM), ω-nitro-L-arginine methyl ester (L-NAME, 10 mM) and sodium pyruvate (NaPyr, 100 µM) and then were left untreated (white bars) or treated with 50 J laser energy (dotted bars). Control (CT) samples were not pre-incubated with antioxidant agents. Data were obtained from different experiments (*n* = 3) performed in triplicate for each experimental condition. Results are expressed as average pg/mL ± standard deviation (S.D.) of h-BD1 or h-BD2 detected via ELISA assay in the cell conditioned medium after 6 h; * *p* < 0.05 compared to unstimulated CT samples; ## *p* < 0.001 compared to 50 J laser-treated CT samples.

**Figure 5 antioxidants-12-01550-f005:**
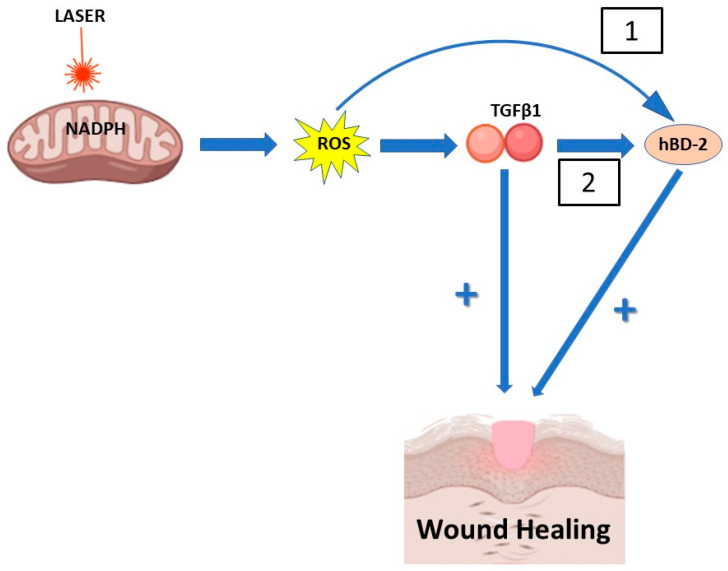
Putative mechanisms of human β-defensin-2 (hBD-2) production induced by laser biostimulation in keratinocyte. Laser light absorption occurs by chromophores in the mitochondria and by other photoacceptors. Emerging evidence supports a role of reactive oxygen species (ROS) induced by laser biostimulation in the induction of the biological effects of this therapy. The ROS could (1) directly increase hBD-2 production or (2) indirectly stimulate TGFβ1 activation, which in turn stimulates hBD-2 production. Thus, TGFβ1 and hBD-2 might act synergistically to accelerate wound healing.

## Data Availability

The data presented in this study are available on request from the corresponding author.

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
