# Peer review of "Laser Biostimulation Induces Wound Healing-Promoter β2-Defensin Expression in Human Keratinocytes via Oxidative Stress"

_antioxidants, 2023, doi:10.3390/antiox12081550_

Round 1

Reviewer 1 Report

Thank you for submitting the manuscript. Before a potential publication there are some comments and improvements to be made to the manuscript, which I have named below.

Line 62: subdivided

Line 84 ff: it is not helpful in this area to talk about three different laser applications within three sentences. The authors speak here of biostimulation, radiation and photobiostimulation. Presumably, these are the same type of laser application. However, this is confusing for the reader and should be changed when referring to the laser application throughout the text. 

Line 133: What LPS strain was used?

Figure 1: three asterisks are not visible in the figure. Please correct. Furthermore, one significance statement as described in the statistical description is sufficient. Furthermore, despite significance in the h-BD1 values, one should think about their meaningfulness in the comparison. The difference is only a few pg. The situation is quite different for h-BD2. Here, intergroup comparisons would also be helpful from the point of view of the expert in order to be able to make appropriate statements. The comparison only with the control group is not sufficient. 

Furthermore, it remains to be discussed to what extent the actually protective mechanism by laser stimulation leads to a comparable reaction compared to stimulation with LPS. 

Therefore, to what extent can we speak of a protective mechanism by laser stimulation when LPS as a negative example achieve the same effect? This is true for both the 6 hours and 24 hours values!

Discussion: the reviewer still takes the discussion at one or the other point in the current result situation clearly too far. It would indeed be more interesting at this point to integrate the further investigations proposed by the authors to improve the scientific basis already in this manuscript. As a prospect, a clear presentation of the possible applications for the laser in daily routine could be considered.

minor spellings

Author Response

Line 62: subdivided    corrected

Line 84 ff: it is not helpful in this area to talk about three different laser applications within three sentences. The authors speak here of biostimulation, radiation and photobiostimulation. Presumably, these are the same type of laser application. However, this is confusing for the reader and should be changed when referring to the laser application throughout the text.

The term “Laser biostimulation” has been used through the entire text

Line 133: What LPS strain was used?

LPS used (cat.n. L5293) was purchased by Sigma-Aldrich (Saint Louis, USA) and obtained from Esche-richia coli 0111:B4.

This sentence has been added to Materials & Methods section.

Figure 1: three asterisks are not visible in the figure. Please correct.

Corrected

Furthermore, one significance statement as described in the statistical description is sufficient.

We belive that is important to show that laser biostimulation effect is more statistically significant in function of  laser intensity used , that is why we indicated in Fig 1 and 2 two levels of significance,

Furthermore, despite significance in the h-BD1 values, one should think about their meaningfulness in the comparison. The difference is only a few pg. The situation is quite different for h-BD2.

 Here, intergroup comparisons would also be helpful from the point of view of the expert in order to be able to make appropriate statements. The comparison only with the control group is not sufficient.

According to the scientific literature, of the two beta defensins studied, only hbd2 appears to be truly inducible. In fact, while laser biostimulation induced a small but significant increase in hBD-1 , hBD-2 was strongly induced.

We showed this difference describing our results not only as pg/ml of hBD-1 and hBD-2 produced but also as a percentage of basal level, instead to indicate an intergroup statistical analysis. We added the following sentences to the results section.

 To underline the marked difference in the ability of the laser biostimulation to induce the two different defensins, it was useful to express the concentrations of hBD-1 and hBD-2 induced as a percentage of the baseline (non-stimulated samples), which have been already indicated as almost equal. Therefore we observed that laser stimulation was able to induce a hBD-1 level variable between 125±8% (5J) and 156±17% (50J) of baseline, while hBD-2 concentration resulted increased in a range between 131 ±17% (5J) and 311±39% (50J) of baseline.

Furthermore, it remains to be discussed to what extent the actually protective mechanism by laser stimulation leads to a comparable reaction compared to stimulation with LPS.

Therefore, to what extent can we speak of a protective mechanism by laser stimulation when LPS as a negative example achieve the same effect? This is true for both the 6 hours and 24 hours values!

The LPS stimulus used in our paper as a positive control is a physiological mechanism that induces the production of both defensins under conditions of bacterial invasion.

We have demonstrated that defensins levels comparable to those induced by LPS can be induced by a single laser biostimulation. The fact of inducing beta defesins production in the absence of bacterial stimulation and under controlled and safe conditions can create a preventive defense in the case, for example, of minor surgical interventions or even in the presence of metabolic dysfunctions (e.g. diabetes).

Therefore, the concept of laser biostimulation “ defensive effect” has to be related to this scenario rather than to the fact that LPS induces an innate immunity response.

We add a small sentence to Dicussion section to try to clarify this point.

Discussion: the reviewer still takes the discussion at one or the other point in the current result situation clearly too far. It would indeed be more interesting at this point to integrate the further investigations proposed by the authors to improve the scientific basis already in this manuscript. As a prospect, a clear presentation of the possible applications for the laser in daily routine could be considered.

In the discussion section, we try to imagine a possible mechanism for laser-induced redox-sensitive hBD-2 induction and a possible role of this defesins in laser biostimulation of wound healing.

We described this possible mechanism on the basis of both experimental results and data from literature, linking the TGF-β1 presence/activation with hBD-2 production. Furthermore we discuss also the future research directions and possible clinical use of our findings in Conclusions.

Comments on the Quality of English Language

minor spellings  corrected

Reviewer 2 Report

In the paper Laser biostimulation induces wound healing-promoter beta2-defensin expression in human keratinocytes via oxidative stress the authors show how the stimulation of HaCaT cells with a concrete laser wavelength is able to stimulate the production of two types of beta-defensins, 1 and 2 with more strength on human beta-defensin 2. Also, they show that the production of these beta-defensins, especially number 2 is highly affected by the presence of antioxidants molecules suggesting an oxidative stress pathway that directs the expression of these molecules.

In general, the paper is well-written and the results are clear. And the results are well explained with coherent conclusions.

Author Response

Thanks for your kind comment.

Reviewer 3 Report

This paper is report that Laser biostimulation induces wound healing-promoter β2 defensin expression in human keratinocytes via oxidative stress. The experiments were done in vitro, using the keratinocyte (HaCaT), and the methodology, results and conclusions are out-of-date ideas.

Comment:

The study reports a major contribution of the laser biostimulation modulating the production of two β-defen-27 sins, hBD-1 and hBD-2 in human keratinocytes. The study has major flaws and the data do not support the conclusions:
Major:
1.The authors use Elisa to identify human β-defensin 1 and human β-defensin 2 production by HaCaT under laser biostimulation. In fact, the α-defensins are a class of immune effector peptides with broad antimicrobial activity against bacteria. Moreover, the conditioned media from irradiated groups has lots of propeptides.  The authors need to use high throughput screening from conditioned medium to convincingly support the conclusions. Protein chip analysis may help.
2. The author only analyzed the laser induced stress response to keratinocytes through oxidative stress blockers, without analyzing the laser induced stress response to keratinocytes at the molecular level.
3. During the process of wound healing, dermal fibroblasts also play a crucial role in this process. Are human β-defensin 1 and human β-defensin 2 derived from fibroblasts or keratinocytes under laser biostimulation. The authors should give an experimental design to distinguish which cell type has a greater impact. The authors should adopt a less biased approach on this topic.

4. Lack of in vivo experiments in research.

5. The research method is too single and simple

 Moderate editing of English language required

Author Response

1.The authors use Elisa to identify human β-defensin 1 and human β-defensin 2 production by HaCaT under laser biostimulation. In fact, the α-defensins are a class of immune effector peptides with broad antimicrobial activity against bacteria. Moreover, the conditioned media from irradiated groups has lots of propeptides.  The authors need to use high throughput screening from conditioned medium to convincingly support the conclusions. Protein chip analysis may help.

We have used a simple but reliable cellular model on which we have accumulated a lot of experience in recent years and we have focused on the production and recovery of only two of the major beta defensins expressed in the epithelia by analyzing their concentration with the help of the most sensitive technique on the market. Nevertheless, the results obtained encourage us to extend our research also towards propeptides using other techniques.

  1. The author only analyzed the laser induced stress response to keratinocytes through oxidative stress blockers, without analyzing the laser induced stress response to keratinocytes at the molecular level.

The laser induced stress response has been widely studied and we cited some of the most important findings in our references section (e.g. ref n.14 and n.16).

  1. During the process of wound healing, dermal fibroblasts also play a crucial role in this process. Are human β-defensin 1 and human β-defensin 2 derived from fibroblasts or keratinocytes under laser biostimulation. The authors should give an experimental design to distinguish which cell type has a greater impact. The authors should adopt a less biased approach on this topic.

We are aware of fibroblast role in wound healing but again this was a first attempt to clarify laser action on keratinocytes.

  1. Lack of in vivo experiments in research.

The in vivo experiments are now ongoing  based on our initial (this paper)  and subsequent data.

  1. The research method is too single and simple

As stated before our paper was an initial attempt to explore anresearch area not still covered.

Comments on the Quality of English Language

 Moderate editing of English language required

We performed a more accurate editing of English language that we hope make our paper more readable.

Reviewer 4 Report

The article "Laser biostimulation induces wound healing-promoter β2-defensin expression in human keratinocytes via oxidative stress." focuses on the contribution of laser biostimulation modulating the production of two β-defen-27 sins, hBD-1 and hBD-2 in human keratinocytes, is well written and the results are clear, but in the discussion section, a clear presentation of the potential applications of the laser in daily routine should be improved and more discussed.

Furthermore, the author analyzed the laser-induced stress response on keratinocytes, but not on dermal fibroblasts, which play a crucial role in this process. It would be interesting to evaluate this aspect as well in order to have a less partial approach on this topic.

Given and resolved these two issues, the article could be published.

Author Response

We thank the reviewer for the positive comments and also for the two questions asked.

1) In the discussion section we have provided guidance on the general use of laser biostimulation (production of analgesia, modulation of the immune response and wound healing and acceleration of tissue regeneration), but we have not provided an introduction to the potential applications of the laser in routine daily because ours is an early stage article from a clinical point of view.

2) We are aware of the role of fibroblasts in wound healing, but this article was a first attempt to clarify the action of the laser on keratinocytes (HaCaT), a simple but reliable cellular model on which we have accumulated a lot of experience in recent years and we focused on the production and recovery of only two of the main beta defensins expressed in the epithelia by analyzing their concentration with the help of the most sensitive technique on the market. Also, so far the only fibroblasts that have been shown to produce h-BD are
limbo-corneal fibroblasts (h-BD1) and human gingival fibroblasts (hBD-2) after bacterial indication (Bautista-Hernández et al., Eur J Microbiol Immunol (Bp). 2017 Sep;7(3):151–157)
However, the obtained results encourage us to extend our research using a 3D printed mucosa-like structure with both keratinocytes and dermal fibroblasts. We are now working on this model with laser biostimulation.

Round 2

Reviewer 1 Report

The authors have somehow corrected the manuscript. Even If I am not ok with everything I think the revision is ok now.

One thing still does not comply with me. Statistical significance cannot be more or less. It is just a number and if there is a significance level of 0.05 it is significant and not more significant reaching 0.001 or anything less. Furthermore why do the authors now use * and # to demonstrate this in their figures. Does not make sense to me.

Author Response

In reference to the p-value issue, I hope I can explain why we have indicated both the p-value < 0.05 and the one < 0.001. Assuming that the p-value is a statistical measure of the likelihood of obtaining the observed outcomes, a p-value of 0.05 or less is generally considered statistically significant. At the same time, however, a p-value much lower than 0.05 indicates a greater probability that the observed phenomenon occurs. That's why, as in many other articles, we wanted to indicate the p value <0.001 and we simply tried to simplify/improve relative graphics.

Reviewer 3 Report

The innovation of this study is average and the selection method is too outdated

No

Author Response

We are happy that the reviewer now considers the quality of our work to be average, but we are sorry that no ratings have been indicated in the comments table other than the one concerning the request for Extensive editing of English language. We would just like to point out that one of the authors (YP) is native English speaker.

Reviewer 4 Report

I thank the authors for their replies and as far as I am concerned the article can now be published